# Mechanisms of Lung and Intestinal Microbiota and Innate Immune Changes Caused by Pathogenic Enterococcus Faecalis Promoting the Development of Pediatric Pneumonia

**DOI:** 10.3390/microorganisms11092203

**Published:** 2023-08-31

**Authors:** Zhiying Tian, Ting Deng, Xuwen Gui, Leilei Wang, Qiulong Yan, Liang Wang

**Affiliations:** 1Stem Cell Clinical Research Center, National Joint Engineering Laboratory, Regenerative Medicine Center, The First Affiliated Hospital of Dalian Medical University, No. 193, Lianhe Road, Shahekou District, Dalian 116011, China; tzy2351805009@163.com; 2Department of Biotechnology, College of Basic Medical Science, Dalian Medical University, Dalian 116044, China; dtings123@163.com (T.D.); guixvwenn@163.com (X.G.); w1446798269@163.com (L.W.); 3Department of Biochemistry and Molecular Biology, College of Basic Medicine, Dalian Medical University, Dalian 116044, China; qiulongy1988@163.com

**Keywords:** *Enterococcus faecium*, flora analysis, immune reaction, pediatric pneumonia, 16srRNA

## Abstract

Bacterial pneumonia is the main cause of illness and death in children under 5 years old. We isolated and cultured pathogenic bacteria *LE* from the intestines of children with pneumonia and replicated the pediatric pneumonia model using an oral gavage bacterial animal model. Interestingly, based on 16srRNA sequencing, we found that the gut and lung microbiota showed the same imbalance trend, which weakened the natural resistance of this area. Further exploration of its mechanism revealed that the disruption of the intestinal mechanical barrier led to the activation of inflammatory factors IL-6 and IL-17, which promoted the recruitment of ILC-3 and the release of IL-17 and IL-22, leading to lung inflammation. The focus of this study is on the premise that the gut and lung microbiota exhibit similar destructive changes, mediating the innate immune response to promote the occurrence of pneumonia and providing a basis for the development and treatment of new drugs for pediatric pneumonia.

## 1. Introduction

Despite improvements in medical care, pediatric pneumonia remains one of the leading causes of death in infants and children under five years of age worldwide [1]. Pneumonia is a common infectious lung condition that causes inflammation, resulting in decreased oxygenation, shortness of breath, and mortality. In 2011, an estimated 1.2 million children under the age of five years died from pneumonia. The vast majority of these fatalities occur in underdeveloped nations where access to treatment is restricted, and efforts to improve care in wealthy countries are lacking [2]. Determining the specificity of the pathogen remains a challenge because of the high mortality rate associated with pediatric pneumonia. Blood cultures have low sensitivity, whereas radiological findings are nonspecific and do not distinguish between viral and bacterial causes of pneumonia [3].

In order to simulate the production of pneumonia caused by *Enterococcus faecalis* entering the intestine, we simplified the process of mouse modeling as follows: Gavage with two species of *Enterococcus faecalis* (Normal *Enterococcus*, *NE* and L-Pathogenic *Enterococcus*, *LE*) of the same genus but different species caused pneumonia symptoms in mice. The treatment of pediatric pneumonia is divided into clinical treatment and pathogen detection. The currently available methods do not allow for the identification of the causative organism in a relatively short time, which increases the mortality rate of the disease. Currently, the main pathogenic mode of bacterial pneumonia is respiratory tract infection, increasing the inflammatory response in the lungs [4,5], and we found that intestinal isolates of *LE* from children with pneumonia that enter the GI tract through gavage can also lead to pulmonary infection. In earlier studies, the gut–lung axis theory was proposed [6], which states that the mammalian gut and lungs have the same developmental starting point during the embryonic period, resulting in a similar physiological structure and protein content. Therefore, we examined the composition of the intestinal and pulmonary flora in mice colonized with *LE* and *NE*.

The lungs and intestines in the body are places where a large number of microorganisms gather [7], and examining their flora structure according to this property can effectively evaluate their interactions [7]. The intestine is a closed cavity that prevents microorganisms from entering other tissues while also ensuring the good reproduction of specific microorganisms in the intestine [8]. On the other hand, intestinal atresia depends on the good expression of some Tight junction proteins, which enables the intestinal tract to use bacteria to decompose nutrients while keeping the abdominal cavity sterile [9]. As the first line of defense of the intestinal tract, tightly connected protein has a powerful role in preventing intestinal microorganisms from entering the abdominal cavity. In addition, studies have confirmed that tightly connected protein ZO-1 can be broken into segments in the mouse intestinal injury model caused by lipopolysaccharide, causing toxins to enter the blood and cause pneumonia [10]. The existing theory regards Tight junction protein as a mechanical barrier of the intestine. If the barrier is destroyed, many diseases will occur, such as sepsis caused by intestinal necrosis, which will lead to multiple organ failure and endanger life [11]. For example, studies have found that octanoic acid can reverse the redistribution of ZO-1 and claudin-1 proteins in LPS-treated Caco-2 cells by increasing cell viability, reducing epithelial permeability, and restoring the expression of claudin-1, ZO-1, and occludin proteins and mRNA, thereby controlling the progression of intestinal barrier injury. For example, studies have found that octanoic acid can reverse the redistribution of ZO-1 and claudin-1 proteins in LPS-treated Caco-2 cells by increasing cell viability, reducing epithelial permeability, and restoring the expression of claudin-1, ZO-1, and occludin proteins and mRNA, thereby controlling the progression of intestinal barrier injury [12]. Some studies have confirmed that not only does the intestinal tract have a mechanical barrier formed by a Tight junction structure, but the lung also has a barrier to isolate microorganisms. This is because the intestines and lungs are in the same embryonic layer during embryonic development and share a developmental starting point [13]. So, lung diseases are closely related to the level of health in the intestines. Pulmonary inflammation is usually associated with elevated levels of inflammatory factors in the blood, and pathological changes may occur in both the intestines and lungs [14]. Serum IL-6 and IL-10 have been suggested to be directly associated with the development of pneumonia symptoms, and their expression levels in the serum can indirectly reflect the development of pneumonia in vivo [15]. The large release of immune factors usually stems from the recruitment of immune cells, and there is a mutual promotion, activation, or restraint relationship between different immune cells, which can cause the body’s inflammatory level to be greatly upregulated and cause tissue damage [16].

The gut and lungs are the main sites for microbial colonization in the body, and the balance of their microbiota can maintain a good state of the body; 16srrna sequencing is a means to comprehensively reflect the state of the microbiota [17]. In the 16S full-length sequence comparison results, if there was 97% sequence similarity, it could be recognized as the same species; therefore, it could be tentatively concluded that an OUT (Operational Taxonomic Unit) consisted of all microorganisms belonging to one species [18]. Alpha diversity was used to analyze the flora species diversity, and the higher the Shannon index, the higher the abundance of the flora [19]. The Chao index represents the number of colonies in the sample; the higher the index, the higher the number of species [17], these indices can effectively assess the balance of the flora. Principal component analysis (PCA) is a non-constrained method of data dimensionality reduction analysis that can be used to study the similarity or dissimilarity of the sample community composition; PCA allows for the observation of differences between individuals or groups [20]. Each point in the graph represents a sample and points with the same color are from the same grouping and distance response sample similarity [21]. Bugbane is a tool for phenotypic characterization of bacterial populations, which focuses on phenotypic prediction, including Gram-positive, Gram-negative, biofilm formation, pathogenicity, mobile components, oxygen demand (including anaerobic, aerobic, and parthenogenic bacteria), and oxidative stress tolerance in seven categories [22]; This helps to demonstrate bacterial properties. In summary, 16srRNA sequencing can utilize multiple indices to evaluate the relationship between the state of the microbiota and diseases.

## 2. Materials and Methods

### 2.1. Strains Identification by 16S Full-Length Sequencing

After *Enterococcus faecium* (*NE* and *LE*) were cultured and amplified, their potential pathogenicity was eliminated. Samples were sent to Sangon Biotech (Shanghai) Co., Ltd. (Shanghai, China) for 16S full-length sequencing. The principle of PICRUSt is to infer the gene function spectrum of a common ancestor based on the measured microbial gene sequence, infer the gene function of other untested species in the database, and construct a gene function prediction for the entire microbial lineage. Download the PICRUSt2 version on the official website and input sequencing data for analysis.

### 2.2. Experimental Design and Animals

Male BALB/c mice (3 weeks old, 20–22 g) with stable health and reduced experimental contingency (environmental and life factors) were purchased from Liaoning Changsheng Biotechnology Co. (Shenyang, China). All mice were housed in the specific pathogen-free Animal Center (SPF level) of Dalian Medical University and fed under specific pathogen-free conditions with free access to sterilized pure water and food. For acclimatization, mice were housed at room temperature (25 °C) for at least one week prior to the start of the experiments. All animal experiments were approved by the Animal Management and Use Committee of Dalian Medical University (approval number AEE22034).

Normally, the bacterial tolerance of the mice organism is ten times higher than that in humans, and most of the bacteria will be depleted in the digestive tract of mice, resulting in much lower pathogenicity; therefore, we extracted the potential virulence factor of *Enterococcus faecalis* and concentrated it. Mice were gavaged at 10 mg/kg once a day for one week, after which the animals were euthanized, and blood, feces, lung, and colon samples were collected The mice were randomly divided into seven groups: untreated control mice (*n* = 56, 7 per group, Each group of eight), Control = NC, NE-SP, NE-LTA, NE-ECP, LE-SP, LE-LTA, and LE-ECP.

Pathogenic *Enterococcus faecium* (*LE*) isolated from the intestines of young children with pneumonia and non-pathogenic *E. faecium* (*NE*) isolated from the intestines of healthy young children were gifted from the Microbiology Teaching and Research Department of the Dalian Medical University. The collection and separation of bacteria used has been approved by the children’s guardian. The following procedures were performed to ensure aseptic conditions.

For the extraction of bacterial surface protein (SPs), bacteria were first grown in a Brain Heart Infusion medium (Haibo Biotech, Qingdao, China) at 37 °C for 16 h. After centrifugation at 3000× *g* × 10 min, the bacteria pellets were washed three times with sterile phosphate-buffered saline (PBS) and resuspended in phosphate-buffered saline (PBS; Seven Biotech, Beijing, China) containing lysozyme (2 g/mL; Vazyme, Nanjing, China). Cell pellets were incubated for 1 h at 37 °C and then centrifuged at 5000× *g* × 10 min, and the precipitates were mixed with 4 mol/mL of lithium chloride solution and incubated for 12 h at 37 °C. After centrifugation, dialysis was performed in PBS at 4 °C for 1 h, and freeze-drying was performed at the end of dialysis. For bacterial Lipoteichoic acid (LTA) extraction, *E. faecium* was incubated in Brain Heart Immersion liquid medium at 37 °C for 16 h until the logarithmic phase was achieved, washed three times with PBS, and mixed with 10 mL sterile PBS after sonication (Toshiba Instruments Co., Tokyo, Japan). The bacteria were then mixed with 65 °C-preheated 90% phenol solution with rapid stirring for 30 min, cooled at 4 °C, and then centrifuged at 3000× *g* × 10 min. The collection of the upper aqueous phase was repeated and mixed with 65 °C sterile ultrapure water three times. DNAse, RNA hydrolase, and protease K (10 mg/mL, Solarbio, Beijing, China) were added and incubated for 45 min to digest nucleic acids and proteins, thereby obtaining pure lipoteichoic acid. The mixture was boiled at 100 °C for 5 min to heat-inactivate the enzyme. Finally, the aqueous phase was placed in a dialysis bag (filter coefficient 15,000 kD, Solarbio, Beijing, China) for dialysis in sterile water to remove residual phenol until the liquid and FeCl_3_ no longer produced a black precipitate. The liquid was removed and freeze-dried, and Ultrapure water was made from Milli-Q Advantage A10 (Millipore, Burlington, MA, USA).

For bacterial Extracellular product (ECP) extraction, *E. faecium* was incubated in a Brain Heart Immersion liquid medium at 37 °C for 16 h until the logarithmic phase was reached. It was then washed three times with sterile PBS and centrifuged at 8000× *g* for 15 min at 4 °C. Sterile PBS was added to the bacterial precipitate to prepare a 1 × 10^6^ CFU/mL bacterial solution. After coating, this bacterial concentration resulted in rapid bacterial growth with minimal impurities. High-pressure cellophane (Solarbio, Beijing, China) was added to a plate containing Enterococcus agar. Cellophane is a semi-permeable membrane with a pore size of less than 100 nm that allows bacteria to contact the culture medium and grow rapidly; however, any bacterial secretions remain on the cellophane, allowing an easy collection. The bacterial solution was seeded on cellophane, and the plates were cultured at 37 °C for 24 h. The cellophane was then washed with sterile precooled PBS. The product was collected and centrifuged (8000× *g*, 4 °C) to retain the supernatant. Any residual *E. faecium* was killed by adding 10 mg/mL phenyllactic acid (Yuanye, Shanghai, China). Dialysis in ultrapure water using a dialysis bag (Sorlarbio, Beijing, China) was conducted to remove residual medium impurities and phenylacetic acid (−4 °C, 48 h) and freeze-dried with polyethylene glycol 20,000.

### 2.3. Hematoxylin-Eosin Tissue Staining

The colons and lungs of mice were collected and fixed in 4% paraformaldehyde (Seven Biotech, Beijing, China). The organs were subjected to ethanol gradient dehydration, liquid paraffin wax wrapping, and wax pourer embedding. Tissue sections were cut using a microtome (Thermo Fisher Scientific, Waltham, MA, USA). Before staining, the tissue samples were dried in an oven (Lichen Technology, Shanghai, China) at 60 °C for 2 h. Tissue slices were dewaxed, stained with hematoxylin and 0.5% eosin (Solarbio, Beijing, China), ethanol gradient-dehydrated, and sealed with neutral gum (Solarbio, Beijing, China).

### 2.4. Immunohistochemistry

Tissue slices were immersed in sodium citrate antigen repair solution (Seven Biotech, Beijing, China) for tissue antigen repair. Samples were incubated with a peroxidase suppressor (Zhongshan Jinqiao, Beijing, China) for 20 min and goat serum (Zhongshan Jinqiao, Beijing, China) for 45 min. Next, the samples were incubated at 4 °C with a goat anti-rabbit Claudin-1/ZO-1 antibody (1:100 dilution; Proteintech, Rosemont, IL, USA), and biotin-labeled goat anti-mouse/rabbit IgG (Zhongshan Jinqiao, Beijing, China) was used as a secondary antibody. The samples were incubated with horseradish enzyme-labeled streptavidin working solution for 20 min. DAB developing solution (Zhongshan Jinqiao, Beijing, China) was prepared to cover the tissues, and a hematoxylin staining solution was used for nuclear staining. Finally, the slices were subjected to gradient dehydration and sealing.

### 2.5. Enzyme-Linked Immunosorbent Assay (ELISA)

IL-6 and IL-10 levels were measured using ELISA kits (Enzyme Biolabs, Jiangsu, China) according to the manufacturer’s instructions. The sensitivity of the mouse IL-6 and IL-10 quantification kits ranged between 3.75 pg/mL–120 pg/mL and 15 pg/mL–400 pg/mL, respectively. GraphPad (version 8.0) was used for the t-test analysis, and the experiments were replicated three times or more.

### 2.6. Quantitative Real-Time Polymerase Chain Reaction 

Total RNA from the mouse lung tissue was extracted using TRIzol reagent (TaKaRa, Japan) according to the manufacturer’s instructions. The RNA quantity and quality were measured in a Nanodrop spectrophotometer (Thermo Fisher Scientific) with a 260/280 ratio of 1.8–2.0. cDNA was reverse-transcribed using a HiScript II Q RT SuperMix qPCR kit (Vazyme, Nanjing, China) according to the manufacturer’s instructions. Gene expression analysis was performed on a Bioer LineGene 9660 Plus system (Bioer, Hangzhou, China) using ChameQ Universal SYBY qPCR Master Mix (Vazyme, Nanjing, China). The relative amounts of each gene were normalized to that of *ACTB* (as a housekeeping gene) and analyzed using the 2^−ΔΔCt^ method. The amplification procedure was the following: pre-denaturation at 95 °C for 30 s, denaturation at 95 °C for 10 s, annealing at 60 °C for 30 s, and extension at 72 °C for 1 min for 40 cycles. 

### 2.7. Western Blot

Total protein was extracted from intestinal tissues or cell lysates using RIPA buffer containing phosphatase inhibitors (Solarbio, Beijing, China), and protein quantification was performed using a Kormas G-250 kit (Sorlarbio, Beijing, China). Protein lysates were separated using 10% sodium dodecyl sulfate-polyacrylamide gel electrophoresis (Beyotime, Beijing, China) and transferred onto polyethylene fluoroethylene membranes (Millipore, Burlington, MA, USA). The membranes were blocked for 30 min at 25 °C with a blocking buffer consisting of 5× fast blocking solution (Beyotime, Beijing, China) in Tris-buffered saline and 0.1% Tween-20. The membranes were then incubated overnight at 4 °C with primary antibodies and horseradish peroxidase-conjugated secondary antibody (Proteintech, Rosemont, IL, USA). Proteins were visualized using an electrochemiluminescence kit (Keygen Biotech, Nanjing, China) and a chemiluminescence imaging system (Bio-Rad, Hercules, CA, USA). The following primary antibodies were used: anti-occludin (OCC) and anti-claudin-1 (CLDN1). The experiments were performed in triplicates. The gray band intensity was calculated using ImageJ, and GraphPad (version 8.0) was used for ANOVA.

### 2.8. Immunofluorescence

Paraffin sections of lung and colon tissue were dewaxed, rehydrated, and subjected to antigen heat retrieval in citrate buffer, followed by blocking with 10% goat serum for 30 min at 25 °C. Next, the samples were incubated with mouse antibodies against RORg^T+^ (1:100 dilution; Bioss, Beijing, China) overnight at 4 °C. After washing three times with PBS, the sections were stained with the corresponding fluorescent secondary antibody (1:100 dilution; Proteintech, Rosemont, IL, USA) for 1 h at 25 °C. The sections were washed three times with PBS and stained with 4′,6-diamidino-2-phenylindole (Seven Biotech, Beijing, China) for 10 min at 25 °C. Images were captured using an inverted fluorescence microscope (Olympus, Shinjuku, Japan). A quantitative method was used to calculate the positive areas in the tissue using the ImageJ 1.8.0 software. The total number of tissue areas was 2, 217, and 984 using the automatic threshold calculation of positive areas. GraphPad software (version 8.0) was used for the ANOVA.

### 2.9. 16SrRNA Sequence 

The DNA from collected mouse feces was isolated according to the instructions of the Bacterial DNA Isolation Kit (Chengdu Fuji Biotechnology Co., Chengdu China), and the 16S V4 region of the bacteria in mouse feces was amplified via PCR and sequenced using the Illumina NovaSeq6000 platform (Illumina, San Diego, CA, USA).

### 2.10. Statistical Analysis

The experiments were repeated in triplicates. Data were analyzed using Prism 8.0 (GraphPad Software, San Diego, CA, USA). Multiple groups were evaluated using one-way analysis of variance (Including Western blotting and immunofluorescence experiments). *p*-value < 0.05 was considered statistically significant.

## 3. Results

### 3.1. 16srRNA Sequence

Since we found that *LE* can still cause pneumonia in mice after intragastric administration, in order to explore the changes in the intestinal and pulmonary flora, we detected the changes in the intestinal and pulmonary flora, respectively. As shown in Figure 1A, there were more intestinal bacterial species than lung bacteria, and the mice in the LE-LTA group had the most species in the intestine and lung. This indicated that mice in the LE-LTA group had more abundant intestinal bacterial species. The three indices of microorganisms in the intestine and lung of mice in the LE-LTA group were significantly lower than those in the control group (Figure 1B, *p* < 0.05), indicating that the diversity and abundance of intestinal and pulmonary flora of mice in this group were considerably lower, thereby the intestinal and pulmonary flora of mice in this group were severely damaged; whereas, no considerable difference in other experimental groups was found. The variability between LE-LTA and the other groups is reflected in Figure 1C, and the points in the figure are evenly distributed in the same circle, indicating good intra-group reproducibility and plausible significance. By comparing the intestinal and pulmonary microbiota to the BUGBASE (Figure 1D), we found that the pulmonary and intestinal microflora showed the most considerable increase in Gram-negative bacteria in the LE-LTA group, accompanied by remarkable enrichment of pathogenic biofilms and an increase in aerobic bacteria, indicating significant similarity between gut and lung microbial communities; The potential toxicity of the LE-LTA group was significantly present in the intestine but did not increase in the lungs, indicating that *LE* plays a pathogenic role in the intestine and may indirectly affect the lungs. The clustering heat map of the sample species composition in Figure 1E shows that the intestine of the mice in the LE-LTA group lacks beneficial bacteria of the genus *Rhodobacter*, and in the lungs, lactobacilli and other beneficial bacteria maintain the balance, whereas the other experimental groups did not differ from the non-experimental groups. Using Picrust2 to predict the metabolism of intestinal and pulmonary microflora (Figure 1F), KEGG results showed that intestinal microflora is very active in the metabolism pathway of ansamycin, which has a guiding role in the antibacterial drug screening. In addition, the pulmonary flora is remarkably sensitive to glycolysis and gluconeogenesis pathways, and some glucose absorption inhibitors are expected to be used in the antibacterial treatment of *LE*. Based on the above experimental results, the abundance of beneficial bacteria in the intestine and lungs of mice in the LE-LTA group decreased. Furthermore, there was an increase in potentially pathogenic bacteria in the intestine but not in the lungs. Therefore, we speculate that LE-LTA can cause pulmonary flora disorders through changes in the intestinal flora, leading to pneumonia.

### 3.2. Pathological Findings of HE Staining of Mice Intestine and Lung

To observe the pathological changes in the intestine and lungs of mice, tissue sections were stained, and it was found that mice in the LE-LTA group had broken intestinal villi and disorganized intestinal gland arrangement compared to the control group, accompanied by a large amount of neutrophil infiltration (Figure 2A), collapsed alveolar cells, fused lung septa, and the presence of inflammatory cells, whereas mice in the *NE* group were not significantly different from those in the control group (Figure 2B). This indicates that both the intestinal tract and lungs of mice in the *LE* group showed obvious damage and inflammatory changes (*p* < 0.05).

### 3.3. Mice Intestinal Closed Junction Protein Assay

The extent of in situ expression of mouse intestinal mucopolysaccharides and the closed junction proteins Claudin-1 and ZO-1, using immunohistochemistry, can probe the integrity of the mouse intestinal barrier more precisely. As shown in Figure 3, there was a considerable decrease in the closed junction proteins in the intestinal tract of mice in LE-LTA group 2 compared to that in the control group, while there was no statistically significant difference in the NE group mice.

### 3.4. ELISA

Inflammatory factors are indicators that can predict the strength of the inflammatory response in vivo. IL-10 and IL-6 are the targets of experimental assays as the first effectors to appear in inflammatory lung infections [23,24]. The results showed that the expression of the two inflammatory factors in the serum, intestine, and lung tissues of mice in the LE-LTA group tended to increase significantly compared to those in the control group (Figure 4, *p* < 0.05), whereas there was no considerable change in the *NE* group, indicating that *LE* can cause a severe inflammatory response in mice.

### 3.5. RT-PCR

Although the inflammatory factor expression levels in the serum fluctuate, the specific site of the large increase remains unexplored. The prognosis and progression of pneumonia are directly related to the high levels of interleukins IL-17 and IL-22 in the lungs [25,26]. Therefore, qPCR was used to detect the expression of these two interleukins in the lung and intestinal tissues. As shown in Figure 5, the expression of IL-17 and IL-22 in the lungs and intestines of mice in the LE-LTA group was much higher than that in the control group (*p* < 0.05), whereas there was no significant increase in the other groups (*p* > 0.05).

### 3.6. Immunofluorescence 

Fluctuations in the expression levels of interleukins IL-22 and IL-17 are closely associated with the pro-inflammatory travel of type 3 intrinsic lymphocytes (ILC-3 cells) in vivo and that ILC-3 cells can be transferred within the mesenteric lymphatics to other tissues in the body [27]. Therefore, we used immunofluorescence to detect the presence and specific location of ILC-3cells in the intestines of mice and found that ILC-3 cells were significantly increased in the intestines of mice in the LE-LTA group and were present in the lymphatic vessels in the field (Figure 6). This indicates that the increase in interleukins IL-22 and IL-17 in this group of mice was significantly associated with ILC-3 cells (*p* < 0.05). Mice in the other groups were not significantly different from those in the control group.

### 3.7. Western Blot

From the results of previous experiments, we found that there was a serious disruption of the intestinal barrier caused by the decreased expression of intestinal closure proteins in mice. The lungs, which are also a tissue for microbial colonization, contain a barrier formed by confined proteins. Using immunoblotting to detect the expression of closure proteins in mouse lungs (Figure 7), we found that the expression of Occlidin and CLDN1 proteins in the lungs of mice in the LE-LTA group significantly decreased, which was consistent with the immunohistochemical results (*p* < 0.05). No statistically significant differences were observed between the other mice and the control group. This indicated that LE-LTA caused an imbalance in the mice’s intestinal and pulmonary flora. The phenotypic characterization of the bugbane flora demonstrated that the intestine flora of the mice in the LE-LTA group tended to be more pathogenic, aerobic, and mostly Gram-negative, whereas the lung flora showed strong biofilm formation, stress tolerance, and aerobics, which may be indicative of the disorder of the intestine and lung flora caused by *LE*.

## 4. Discussion

Pediatric pneumonia, a disease affecting infants and children with the highest mortality rate in developing countries, affects the survival and growth of children under five years of age [28,29]. Existing studies have focused on respiratory tract infections [30]. Previous studies have found that lung diseases can be caused by changes in intestinal function and imbalanced microbiota, thus validating the gut–lung axis hypothesis [31]. The intestine has been considered the main digestive site in the past, and more than 70% of the body’s immune cells are found in the mucosal layer of the intestine [32,33], where they live in symbiosis with hundreds of millions of microorganisms that form adaptive immunity [34,35]. The presence of a large number of immune cells in the intestine is due to the structure of the intestine [36], which has countless intestinal villi and a large number of crypt foci, both of which support the presence of immune cells [37,38]. As medical research progresses, it is increasingly being recognized that alterations in the intestine physicochemical environment are inextricably linked to the development of lung diseases [39]. Intestines and lungs are very similar in their embryonic development, creating a similar physiological basis for both [40], such as the presence of microbial colonization, large number of immune cells in the mucosa, and involvement in the formation of the acquired immune memory of the body [41,42]. Using pathogenic *Enterococcus faecalis* isolated from the intestines of pediatric patients with pneumonia, the same bacterial genus was used for comparison with normal human *Enterococcus faecalis* to investigate why the disease manifests differently.

In recent years, the pathogenesis of various human diseases has been attributed to an increase in intestinal permeability, although scientific evidence supporting this hypothesis has been weak. However, in the past decade of research, an increasing number of researchers have focused on human genetics, gut microbiome, and proteomics, demonstrating that the loss of gut mucosal barrier function, especially the loss of gut mucosal barrier function, may seriously affect antigen transport and ultimately lead to chronic inflammation in genetically susceptible individuals, including autoimmune diseases [43]. Western blotting showed that the expression of the Tight junction proteins CLDN1 and Occlidin, which form the basis of the intestinal mechanical barrier in mice [44,45], was considerably decreased in the lung and intestine tissues of the LE-LTA group, and the immunohistochemistry results were consistent with the Western blotting results. ELISA results for the inflammatory factors IL-6 and IL-10 in mice showed that pathogenic LE-LTA treatment did. The results of ELISA for IL-6 and IL-10 in mice showed that *E. faecalis* lipophosphatidic acid was able to increase lysophosphatidic in these two inflammatory factors, while the other experimental groups did not show considerably significant results. As the first inflammatory factors in pneumonia, interleukin 6 and 10 can rapidly respond to inflammatory recruitment and mediate immune Cell migration [46]. This indicates that *E. faecalis* lipophosphatidic acid can induce a systemic inflammatory response in the organism. The results of RT-PCR in mouse lungs showed that IL-17 and IL-22 were also remarkably expressed in the lungs of mice, indicating that LE-LTA activates immune cells and thus produces a large number of inflammatory factors; Interleukins 22 and 17 usually activate innate immune cells(ILC) such as ILC-2 and ILC-3, which will make the duration of the immune stress process longer and be accompanied by long-term Immunological memory reprogramming [47,48].

The results of flora structure analysis showed that the relative abundance of beneficial bacteria that participated in the decomposition of nutrients and had anti-inflammatory effects in the intestine of mice in the *E. faecalis* lipophosphoric acid group decreased considerably, and the relative abundance of some harmful bacteria that could cause diseases increased remarkably, suggesting that *E. faecalis* lipophosphoric acid could change the balance in the intestine, disrupt the structure of mice intestinal flora and affect the homeostasis in the intestine [49]. It is worth noting that in the analysis of microbiota structure, the proportion of *Bacteroidea*, *Prevobacteriaceae*, and *Rumen* microbiota in the gut of LE-LTA group mice showed a decreasing trend, while the proportion of Bacillus increased significantly. In recent studies, it has been pointed out by OTU that *Bacteroidea* can promote the decomposition and absorption of intestinal nutrients, thereby balancing the gut microenvironment [47]. However, Bacillus subtilis can release various intestinal toxins, causing various diseases and dysbiosis of the bacterial community structure [50]. In addition, as the cornerstone of the gut, rumen microbiota species play an irreplaceable role in the breakdown of short-chain fatty acids in the gut. They can convert essential fatty acids in the gut and provide a favorable environment for the colonization of probiotics in the gut [51]. Changes in gut microbiota may lead to changes in internal homeostasis that hinder immune cells from maintaining activity [52], so gut microbiota homeostasis plays an important role in stabilizing the body’s immune balance. So, the above results indicate that alterations in the intestinal flora may contribute to changes in internal homeostasis, making it unfavorable for immune cells to remain active.

The combined analysis of the above results showed that LE-LTA disrupted the balance of intestinal flora, decreased the abundance of beneficial bacteria, and considerably increased the proportion of pathogenic bacteria, leading to serious disruption of the intestinal barrier in mice. Furthermore, the toxin enters the bloodstream, leading to the activation of systemic immune responses in mice, resulting in the activation of ILC-3 cells in the colon to produce IL-17 and IL-22 to mediate the inflammatory response [53].

The purpose of this study was to compare changes in intestinal and pulmonary flora and immune indicators between children with pneumonia and healthy children, with the aim of determining the pathogenesis of pneumonia in children. The results of 16SrRNA showed that the intestinal flora, but not the pulmonary, of children with pneumonia (*LE*) in normal children (*NE*) was considerably disordered, with a decrease in beneficial bacteria abundance and an increase in the harmful bacteria abundance; this verifies the conclusion that the imbalance of Gut microbiota will lead to lung disease [54]. To determine the specific mechanisms underlying LE-induced pneumonia in children, various biochemical indicators were tested. We found that the expression of intestinal Tight junction proteins in the LE-LTA group of mice decreased, and the levels of IL-6 and IL-10 in the blood remarkably increased. Innate immune cells in the intestine ILC-3 proliferated considerably, further confirming that the increase in interleukin 6 and 10 can lead to the recruitment of ILC-3 and further lead to the release of IL-17 and IL-22, which mediate inflammation [55]. Because of the upstream and downstream regulatory relationship between IL-17, IL-22, and ILC-3, the expression levels of IL-17 and IL-22 were detected via RT-PCR in the lungs of LE-LTA group mice. It was found that two kinds of interleukins were considerably expressed in the lungs of this group of mice, which further led to lung inflammation. To our knowledge, this is the first study to sequence bacteria in the intestines and lungs of healthy children and children with pneumonia [56]. By comparing the differences, it was concluded that lung inflammation in children with pneumonia is not caused by the destruction of lung bacteria but by the decrease in intestinal Tight junction expression caused by the disorder of intestinal bacteria, which causes the entry of pathogenic enteroglobulin phosphoteichoic acid (LE-LTA) into the lungs to activate ILC-3 cells, resulting in high expression of IL-17 and IL-22, leading to pneumonia. These results not only validated the intestinal lung axis theory using basic experiments on *LE* and *NE* isolated from the intestines of children with pneumonia and healthy children but also provided new directions for the clinical treatment of pediatric pneumonia. We combined the clinical use of *LE* and *NE* isolated from the intestines and lungs of children with pneumonia and healthy children to model mice. After using big data analysis to discover the connection between the lungs and intestines, we further conducted molecular and biochemical methods to discover the pneumonia mechanism triggered by LE-LTA-induced intestinal barrier damage. However, after the above argument, we still need to point OTU the shortcomings of this study. Firstly, this is a special pneumonia model, and the conclusions currently drawn in mice cannot be replicated in humans, so the experimental results have limitations. On the other hand, we have only found that LE-LTA leads to a pro-inflammatory effect on pneumonia, and whether it is a secondary or initial cause of the disease is not yet known. We will proceed with experimental exploration here in the future (The same modifications as the reviewer’s comments). Based on the above experiments, we hope that the Targeted therapy method for infantile pneumonia caused by Enterococcus infection can be changed to promote the expression of ZO-1 and CLDN1 in the intestinal Tight junction while inhibiting the levels of inflammatory factors IL-6, IL-10, IL-17 and IL-22 to achieve a better cure rate.

## Figures and Tables

**Figure 1 microorganisms-11-02203-f001:**
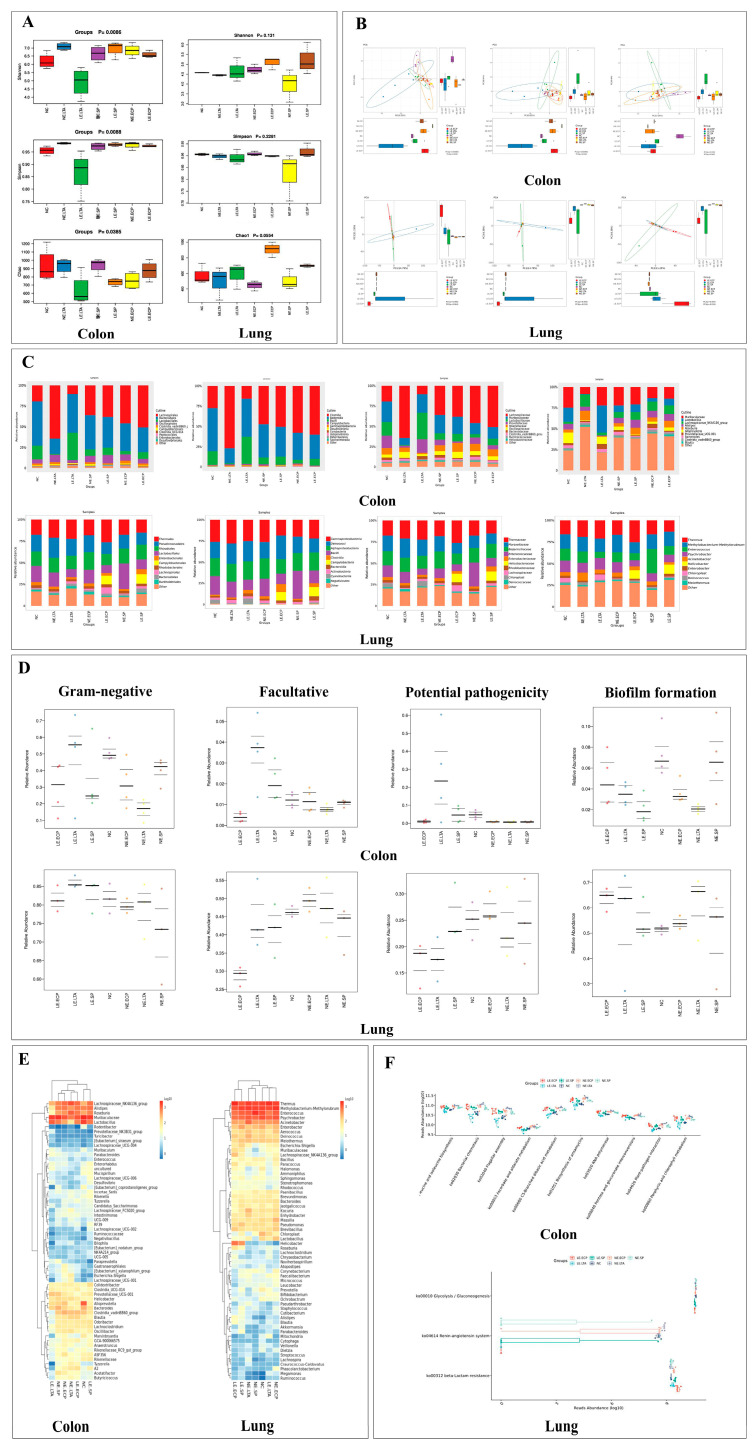
16srRNA sequence of mouse intestinal and lung flora. (**A**) shows the α diversity of mouse intestinal and lung flora. (**B**) showed the results of Beta diversity and PCA of microflora. (**C**) shows the relative abundance histogram of the microbial population. The (**D**) is the analysis of bacterial phenotypic characteristics. (**E**) shows the clustering heat map of species composition. The (**F**) is the KEGG analysis of bacterial flora.

**Figure 2 microorganisms-11-02203-f002:**
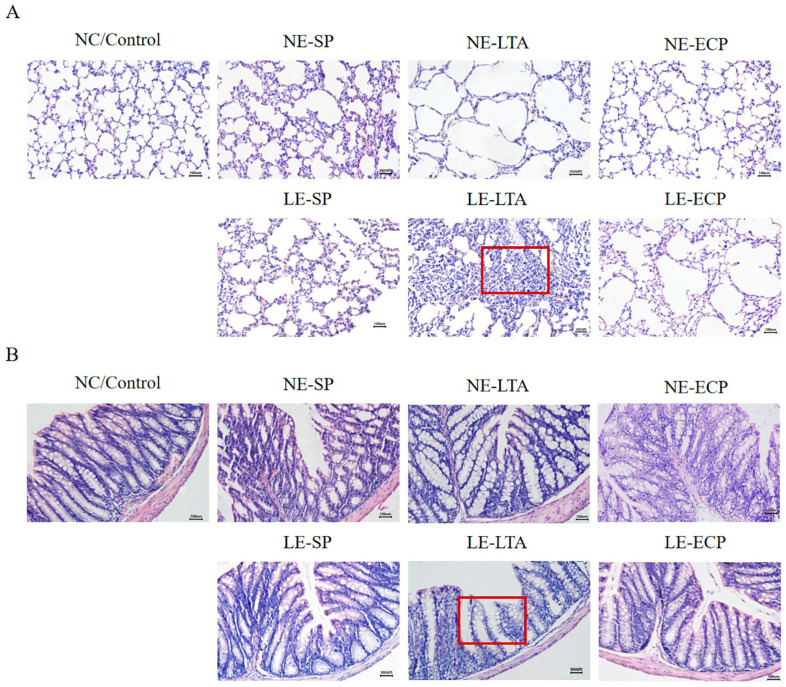
(**A**,**B**) Haematoxylin-eosin staining of lung and intestine tissue samples. Red boxes indicate A collapsed alveolar cells, fused lung septa, and inflammatory cell infiltration, as well as B broken intestinal villi and disorganized intestinal gland arrangement. Scale bar: 200× magnification.

**Figure 3 microorganisms-11-02203-f003:**
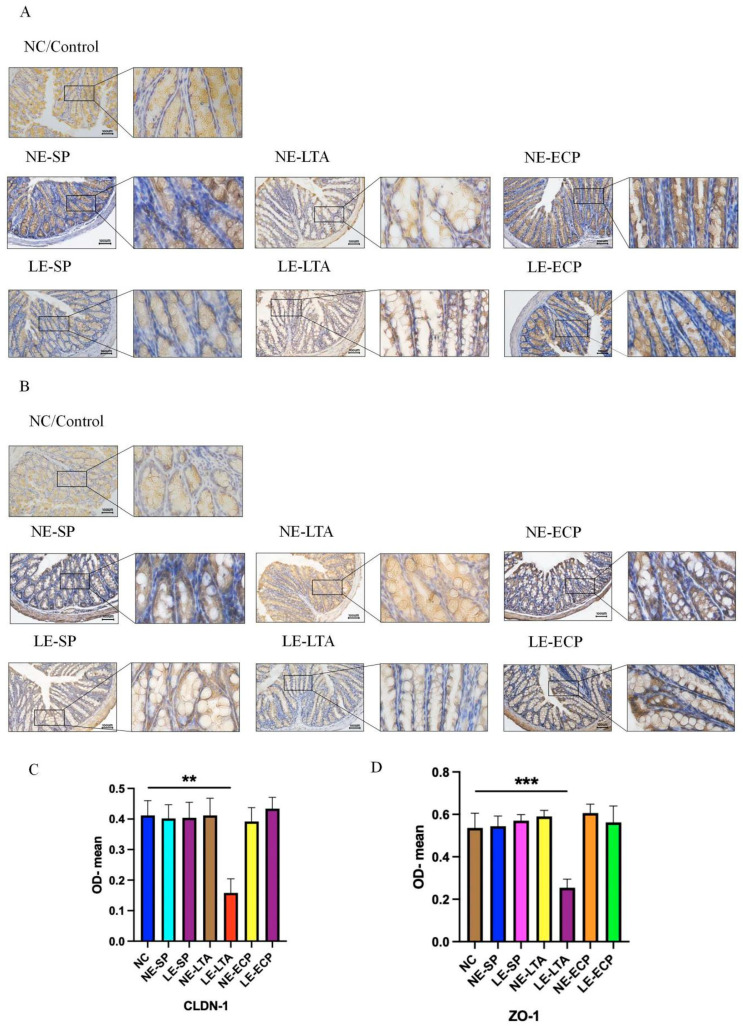
Immunohistochemical detection of Claudin in the (**A**) intestine and ZO-1 in the (**B**) intestine of mice. Quantification of CLDN-1 (**C**) levels and ZO-1 (**D**) levels in the organs using ImageJ (https://imagej.nih.gov/ij/, accessed on 27 March 2022). ** *p* < 0.01, *** *p* < 0.001 for the LE-LTA group compared with the Control.

**Figure 4 microorganisms-11-02203-f004:**
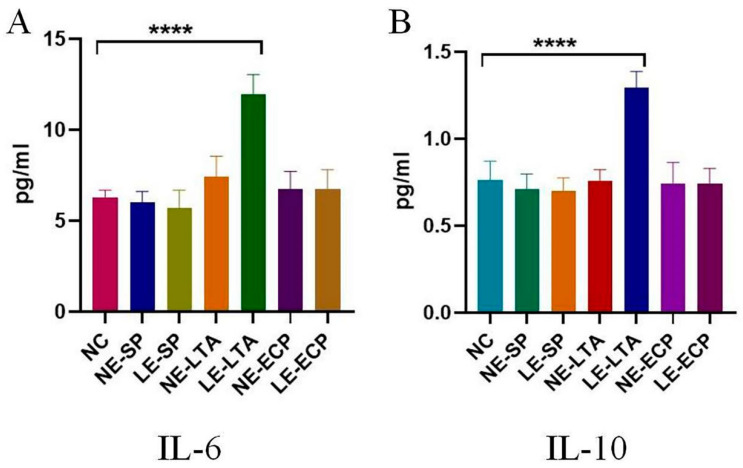
Quantitative analysis of mouse pro-inflammatory factor (**A**) IL-6 and anti-inflammatory factor (**B**) IL-10. **** *p* < 0.0001.

**Figure 5 microorganisms-11-02203-f005:**
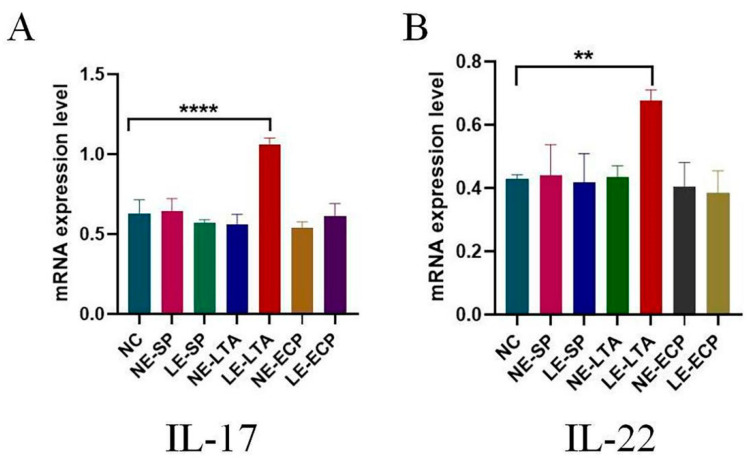
Relative mRNA expression of inflammatory factors (**A**) IL-17 and (**B**) IL-22 in mice. ** *p* < 0.01 and **** *p* < 0.0001. The expression of IL-17 and IL-22 in mice in the LE-LTA group was much higher than that in the control group (*p* < 0.05).

**Figure 6 microorganisms-11-02203-f006:**
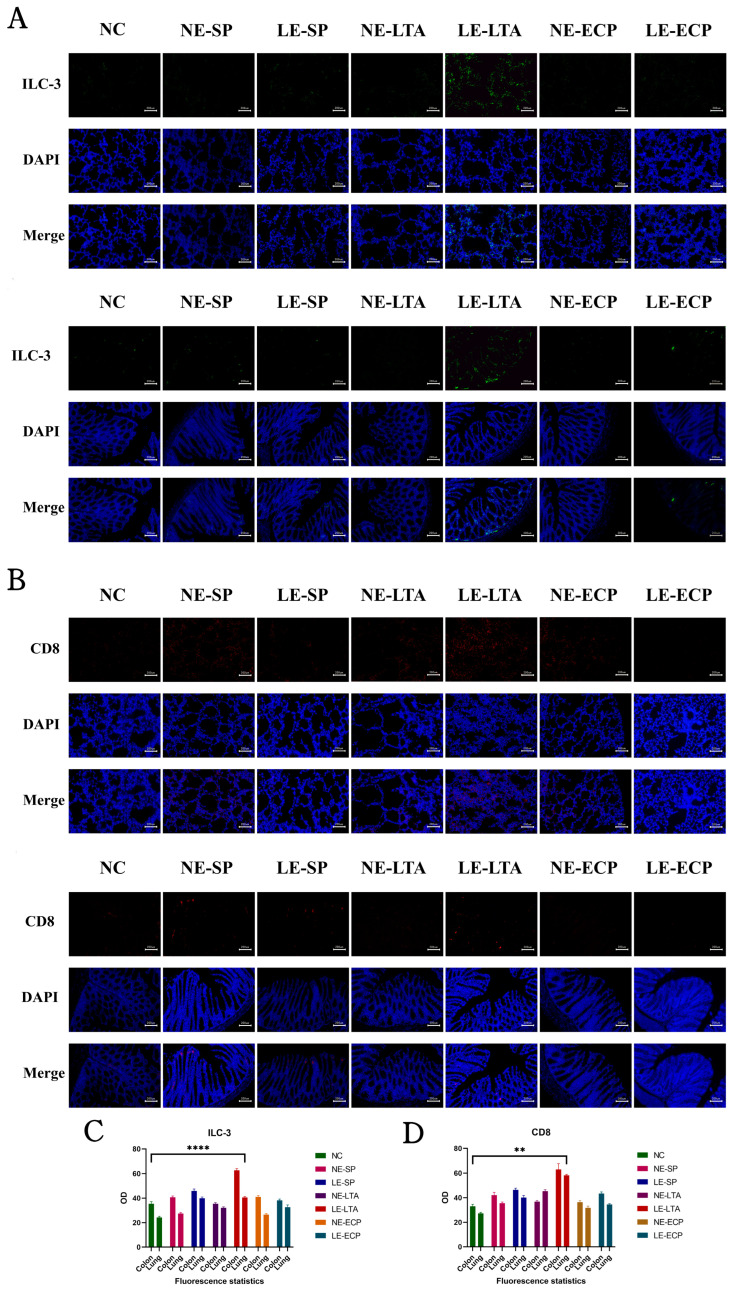
Immunofluorescence detection of ILC-3 (**A**) and CD8+ T (**B**) cells in mouse lung and intestinal tissues. Quantification of ILC-3 (**C**) levels and CD8 (**D**) levels in the organs using ImageJ (https://imagej.nih.gov/ij/, accessed on 27 March 2022). ** *p* < 0.01, **** *p* < 0.0001 for the LE-LTA group compared with the Control.

**Figure 7 microorganisms-11-02203-f007:**
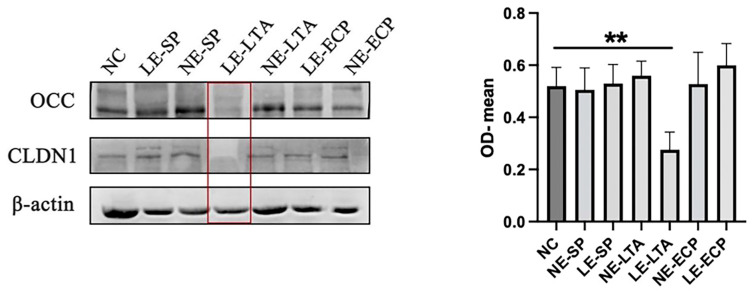
Western blot was used to detect the expression levels of Tight junction protein (Occ and Cldn1) in intestinal homogenates. ** *p* < 0.01.

## Data Availability

Data supporting the findings of this study are available from the corresponding author upon request.

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
