# Peer review of "Mechanisms of Lung and Intestinal Microbiota and Innate Immune Changes Caused by Pathogenic Enterococcus Faecalis Promoting the Development of Pediatric Pneumonia"

_microorganisms, 2023, doi:10.3390/microorganisms11092203_

Round 1

Reviewer 1 Report

In this paper authors replicated a pneumonia model using an oral gavage bacterial animal model. Using 16srRNA sequencing, they found an imbalanced trend  in the gut and lung microbiota.

The paper is novel and intriguing. 

Few notes to improve it: 

- abstract LE is not specified further and it is confusing for the reader. Whenever an acronym appears for the first time it should be explained. 

- introduction papers such as PMID: 23564505 and PMID: 31076989 should be cited and put into context/background frame

- discussion: limitations regarding the generalizability should be acknowledged. It is a very specific and extreme case animal model (very far from bedside!) and conclusions should be not overemphasized 

Author Response

Dear Reviewer: Thank you very much for reviewing my manuscript. As requested, I have made corresponding corrections (in purplefont,Green font is common opinion) in the latest revised manuscript. The details are as follows:

(1) I annotated the first occurrence of the fungal genus on lines 41 to 42.

(2) The background references can be found in lines 59-63, 66-72, and 377-383.

(3) In the discussion section,excessive emphasis waseliminated and the limitations and shortcomings of the research were pointed out.

Finally, I hope my changes can meet the requirements. I am very grateful for your work.

Reviewer 2 Report

 COMMENTS TO AUTHORS

1.  Appropriateness and accuracy of the abstract.

The abstract in its present form is appropriate for the manuscript. After revision of this paper, the authors should look if the abstract needs correction too.

2.  General comments.

The authors present an interesting study about the gut-lung axis theory. The authors underline this gut-lung axis theory by studying the effect of gut inflammation and its impact on inflammatory components in the lung. Although the study is novel and essential for the field, it needs major revision before it can be accepted for publication.

3.  Specific comments and recommendations for revision.

A.    Major

1.     In this manuscript, the authors state that the gut and lung have a common embryologic origin. That is true, but they say it three-time but using different references. Why do they need three various references? Is it not easier to use one? The authors should explain this.

2.     What is the goal of this study? This should be introduced and discussed in Introduction, but not earlier than lines 425-427 this should be mentioned. The authors should do this in the Introduction section rather than the Discussion section.

3.     Line 102: how was their potential pathogenicity eliminated? This should be included.

4.     Line 105: why did the authors choose this strain of mice? They should explain this.

5.     Line 115: how did the authors extract the potential virulence factor? This should be included.

6.     It is still unclear whether the LE bacteria of children were used. As I read in lines 113-120, LE from mice was used. But the next section talks about LE from children. The authors should clearly explain where LE came from. These paragraphs need to be rewritten.

7.     Lines 233-232: the authors used t-tests and ANOVA. This suggests that their data are normally distributed. Was this the case? What test was used to test for normality? This must be included in 2.10.

8.     Line 264: the authors tell the reader that for the first time, they used Picrust2. They should introduce this in the Methods section and explain what it does, the version used, and who the manufacturer is. The same is true for KEGG, mentioned in lines 265 and 280.

9.     Lines 425: from here, the authors are linking children. But can the authors do this? The mice got an extracted amount of LE. How does this relate to exposure in children? Are the results from mice, or certain strains of mice, direct comparable to humans or, in this case, children? It might be, but this should be discussed.

10.  I miss the discussion about the strengths and limitations of this study. The authors must include this in the Discussion section. 

11.  In this edition of the manuscript, figure 1 is hard to read. Please use different figures so the readability is better.

B.  Minor  

1.     Lines 41-43: This sentence is coming out of the blue. What do the authors want to state with this sentence? They need to link it more to the information written before and after this sentence.

2.     Line 54: the authors suggest that lungs are porous. From my point of view, this is not true. Small molecules like oxygen and carbon dioxide can diffuse through the alveolar cells. However, larger molecules and ions need an active medium, like channels. Can the authors explain why they think that the lung is porous?

3.     Line 59 and further: tight junctions are written as “tight junctions” and not as “Tight junctions”.

4.     Line 82: OUT is mentioned for the first time here. The authors should write it in full before abbreviating it.

5.     Line 87: the abbreviation of Principal Component Analysis is PCA and not PCoA.

6.     Lines 109 and 141: add the manufacturer of the sterilized pure water and ultra-pure water

7.     Lines 119, 127, 135, and 148: the authors use many abbreviations, not explaining them before. Only at the end of this manuscript a list of abbreviations is included. I suggest also explaining these abbreviations in the main text.

8.     Lines 186-187: This is part of the statistical analysis and should be included in 2.10

9.     Lines 240-242: I am trying to figure out what the authors want to tell the reader. The authors should rewrite this sentence.

10.  Line 391: ELISA is explained in line 182 and does not need any explanation in this section. Enzyme-linked immunosorbent assay can be deleted.

11.  Lines 401-402: Interleukins 22 and 17 usually activate innate immune cells, such as innate immune cells. I think this is a typo. The authors should look at it and rewrite the sentence.

12.  Line 410-414: the authors lost me. What do they want to tell us? The authors should rewrite this sentence.

The use of the English language and how it is written sometimes needs to be clarified. I suggest that a native-speaking translator edit this paper.

Author Response

Dear Reviewer: Thank you very much for reviewing my manuscript. As requested, I have made corresponding corrections in the latest revised manuscript (in red font, Green font is common opinion). The specific reasons for citing three references are as follows:

(1) The first reason is that there are different emphasis points in the citation explanations, which can better connect with the context. Secondly, it is to avoid the repetition of references.

(2) The animals in line 102 of the article objective.

(3) that have been deleted from the discussion are SPF level mice, as indicated in the manuscript.

(4) The reason for the selection of mice in line 105 has been stated in line 106 as being healthy and stable, with a decrease in experimental chance.

(5) Label the extraction method for virulence factors in lines 126/133/145.

(6) In lines 121 to 123 of the manuscript, it is stated that both NE and LE bacteria were isolated from clinically ill and healthy children, not from mice. We isolated the bacteria from the human body and gavage them to mice to obtain disease models for experimentation.

(7) The t-test in statistics has been removed from the methods section, and this study was conducted using analysis of variance, which is a writing error.

(8) Corrected in the method section.

(9) The description in the manuscript has been changed, as it was obtained from clinical bacteria for mouse model replication, thus contacting children.

(10) The advantages and limitations of the revised research have already been discussed.

(11) Revised the order of image placement and article description in Figure 1.

(12) The introduction of narrative has been added to enhance good readability.

(13) The narrative error has been corrected.

(14) The full text has been changed.

(15) The full name has been added.

(16) Full text changes have been made.

(17) Added in the method section.

(18) It has been added in the text. (19) It has been explained in the method statistics section.

(20) The narrative has been changed.

(21) Enzyme linked immunosorbent assay has been removed from the discussion section.

(22) Writing errors have been corrected.

(23) The description and citation of relevant literature that have been modified with unclear semantics.

Finally, I hope my changes can satisfy you. I am very grateful for your work.
